# Evaluation of a Set of miRNAs in 26 Cases of Fatal Traumatic Brain Injuries

**DOI:** 10.3390/ijms241310836

**Published:** 2023-06-29

**Authors:** Serena Bonin, Stefano D’Errico, Caterina Medeot, Carlo Moreschi, Solange Sorçaburu Ciglieri, Michela Peruch, Monica Concato, Eros Azzalini, Carlo Previderè, Paolo Fattorini

**Affiliations:** 1DSM—Department of Medical Sciences, University of Trieste, 34149 Trieste, Italy; sbonin@units.it (S.B.); caterina.medeot@phd.units.it (C.M.); soleburu@hotmail.it (S.S.C.); michela.peruch@studenti.units.it (M.P.); monica.concato@studenti.units.it (M.C.); eazzalini@units.it (E.A.); fattorin@units.it (P.F.); 2DAME—Department of Medical Area, University of Udine, 33100 Udine, Italy; carlo.moreschi@uniud.it; 3Department of Public Health, Experimental, and Forensic Medicine, Section of Legal Medicine and Forensic Sciences, University of Pavia, 27100 Pavia, Italy; previde@unipv.it

**Keywords:** traumatic brain injuries, miRNA, FFPE tissues, RNA degradation, forensic autopsy

## Abstract

In forensic medicine, identifying novel biomarkers for use as diagnostic tools to ascertain causes of death is challenging because of sample degradation. To that aim, a cohort (*n* = 26) of fatal traumatic brain injuries (TBIs) were tested for three candidate miRNAs (namely, miR-124-3p, miR-138-5p, and miR144-3p). For each case, three FFPE specimens (coup area (CA), contrecoup area (CCA), and the corpus callosum (CC)) were investigated, whereas the FFPE brain tissues of 45 subjects (deceased due to acute cardiovascular events) were used as controls. Relative quantification via the ∆∆Ct method returned significantly higher expression levels of the three candidate miRNAs (*p* < 0.01) in the TBI cases. No difference was detected in the expression levels of any miRNA investigated in the study among the CA, CCA, and CC. Furthermore, the analyzed miRNAs were unrelated to the TBI samples’ post-mortem intervals (PMIs). On the contrary, *has-miR-124-3p* ahas*hsa-miR-144-3p* were significantly correlated (*p* < 0.01) with the agonal time in TBI deaths. Since the RNA was highly degraded in autoptic FFPE tissues, it was impossible to analyze the mRNA targets of the miRNAs investigated in the present study, highlighting the necessity of standardizing pre-analytical processes even for autopsy tissues.

## 1. Introduction

Traumatic brain injuries (TBIs) are the leading cause of traumatic death worldwide [1]. Road accidents falls, and assaults can cause TBIs, with road accidents causing more than 30% of TBIs. TBIs are common under the age of nine or above 80 years of age, while for road accidents, the most affected age range is between 40 and 60 years of age. A TBI is “an alteration of brain function caused by an external force” [2]. TBIs can be classified into primary and secondary brain damage, respectively, depending on whether the TBI is due to direct mechanical injury (fractures of the skull, surface contusions of the brain, axonal injury, and intracranial hemorrhage) or follows biochemical changes (excitotoxicity, free radical generation, neuroinflammatory response, and cell death) [3]. In other words, while the primary damage includes all injuries occurring at the time of the impact with the resulting local neuronal destruction, secondary damage implies multiple signaling cascades that begin immediately after the first moment, leading to neuronal and astroglia injuries [3]. 

The evaluation of TBIs is of great relevance in forensic medicine because TBIs cause death even if no or minimal visible signs are recorded or in the case of decomposed bodies [3]. In addition, the timing of the TBI can also be of interest in forensics [3,4]. To that aim, several approaches are usually employed: post-mortem radiology, autopsy, histology, and immunohistochemistry [3,4,5]. 

In recent years, RNA-based biomarkers, mostly miRNAs, have been proposed as valuable tools for investigating the cause of death [6,7], the post-mortem interval [8], and other topics [9] in forensic medicine [10]. However, the traumatic mechanisms and consequences for the brain are not fully understood at the molecular level. Therefore, identifying novel biomarkers would be extremely interesting for forensic medicine. Previous studies on limited samples have suggested microRNAs (miRNAs) as potential candidate markers in forensic science [11]. 

The central nervous system (CNS) has the highest concentration and diversity of miRNAs. The expression of miRNAs expands during neurodevelopment and varies throughout different brain areas. Their functions are peculiar, as miRNAs regulate local protein expression, synapse maturation, and neuronal circuit formation. A few studies have shown that the levels of certain miRNAs change after trauma [12,13], suggesting a possible role of such molecules in TBIs. Micro-RNAs seem to be released in the extracellular environment by damaged neural through exosomes and/or microvesicles [14]. Nevertheless, most studies focused on murine and rat models for which FFPE (formalin-fixed and paraffin-embedded) brain tissues tissue were easily available [15,16,17]. For humans, miRNAs associated with TBI were instead evaluated mainly in fluids such as peripheral blood, saliva, or cerebrospinal fluid [10,18]. 

According to previous reports in mouse FFPE [15,16,17] brains, a set of seven miRNAs were initially tested in this study. In a second step, three miRNAs were selected as candidate biomarkers (*hsa-miR-124-3p*, *hsa-miR-138-5p*, and *hsa-miR144-3p*) [15,16,17]. MicroRNA-124-3p is the most abundant in the brain. It plays a critical role in neuronal differentiation, neuro-immunity, synaptic plasticity, and axonal growth [19], while *hsa-miR-138-5p* has been implicated in the hypoxia/angiogenesis pathway in glioblastomas [20]. Lastly, *hsa-miR144-3p* has been reported to play a role in intracerebral hemorrhage, neuroinflammation, and neuro-regeneration [21]. Accordingly, this study aimed to investigate the expression levels of the three aforementioned miRNAs in a cohort of 26 cases of fatal TBI and controls.

## 2. Results

In this study, the profiling of three miRNAs was investigated in a cohort of routine autoptic FFPE tissues to possibly define the causes of death and other parameters of forensic interest. The cohort of samples included 62 control FFPE specimens belonging to 45 subjects and 78 FFPE specimens belonging to 26 subjects deceased from brain trauma. The distribution of genders and age at death did not differ between cases and controls (*p* = 1.0 and *p* = 0.2, respectively). In detail, 22 males and 4 females were enrolled in cases, while 38 males and 7 females were included in the control group. The mean age at death was 58 (range: 23–90) years and 55 (range: 18–88) years in the cases and the controls, respectively (see Appendix A).

Multiple specimens were gathered from cases from different brain sites, namely, the coup area, the contrecoup, and the corpus callosum. Each RNA extract was investigated for its quality and the relative expression levels of the three miRNAs, namely, *hsa-miR-124-3p*, *hsa-miR-138-5p*, *hsa-miR-144-3p*.

### 2.1. RNA Quality

A total RNA analysis by Agilent Bioanalyzer returned a median RIN value of 2.40 (range: 1–5.3). In 21 out of 140 samples, the instrument did not return any value, although the DV_100_ and DV_200_ could be calculated. On average, the RIN value did not differ between cases and controls (*p* = 0.1). The DV_100_ value was measured in all extracts with a median value of 64%, and comparable values were found between cases and controls (62.5% in cases and 66% in controls; *p* = 0.09). On the contrary, the DV_200_ value significantly differed between cases and controls, with a considerably higher percentage of longer RNA fragments in controls compared to cases (31% vs. 25%, *p* < 0.001), as shown in Figure 1. 

RNA quality was significantly influenced by the duration of the fixation time, namely, longer or shorter/equal to 48 h. Both the DV_100_ and DV_200_ values were significantly lower in samples with longer fixation times (Figure 2).

### 2.2. miRNA Profiling in Cases and Controls

Taking the low RNA quality assessed via DV_200_, our study focused on the analysis of miRNA, as biased results are expected for mRNA assays because of the enhanced degradation of RNA samples [22]. The expression levels of the analyzed miRNAs, namely, *has-miR-124-3hashsa-miR-138-5p*hasnd *hsa-miR-144-3p*, were significantly different between the cases and controls, with higher expression levels in the cases than the controls, as shown in Figure 3. 

In addition, those miRNAs did not differ significantly among the different brain sites in cases, as reported in Figure 4, with similar expression levels in the trauma area and the corpus callosum and knockback area. This result means that the expression levels of the analyzed miRNAs are independent of the sampling sites in injured brains. 

The results of the logistic regression investigating the relationship among miRNA expression levels, age of death, and year of sample collection in the entire cohort of samples and controls are reported in Table 1. Although the model was significant (*p* < 0.0001), only the expression level of *hsa-miR-138-5p* positively associated with brain traumatism, even in the multivariate analysis. A higher level of expression of *hsa-miR-138-5p* is possibly related to a TBI cause of death. Of note, the logistic regression results showed that the cause of death (outcome in the logistic regression) was not influenced by the year of sample collection or the age at death.

### 2.3. Time Variables and RNA Measures in TBI Deaths

The agonal time, which was recorded in 25 out of 26 TBI deaths (96.2%), had a mean value of 63 h (median value = 3 h; range 0–504). In TBI deaths, the post-mortem interval had a mean value of 4 days (median value = 3 days; range = 1–21). The PMI did not differ significantly between cases and controls (*p* = 0.4). The expression levels of the investigated miRNAs in the traumatic tissues and the indicators of RNA quality, namely, the DV_100_ and DV_200_, were related to the agonal time and PMI via Spearman’s rank correlation, as reported in Table 2. 

In detail, the agonal time did not seem to be related to RNA degradation indices such as the DV_100_ and DV_200_. The agonal times in TBI deaths were significantly associated with the expression levels of two miRNAs. In particular, the duration of the agonal state in TBI was directly associated with the expression levels of *hsa-miR-124-3p* but inversely with *hsa-miR-144-3p.* Conversely, the post-mortem interval, in which biomolecule degradation is expected, was inversely related to the RNA quality index DV_200_. No associations were detected between the post-mortem interval and the expression levels of the analyzed miRNAs.

## 3. Discussion

In TBI, miRNAs are primarily studied in the plasma/serum of patients to assess the trauma’s severity for clinical purposes [10]. In contrast, there is a lack of data on miRNA levels within the brain tissues of deceased subjects [10,12,13]. In this study, after a pre-evaluation of seven miRNAs, the expression levels of three miRNAs were investigated in a set of injured human brains. In particular, *hsa-miR-124-3p*, *hsa-miR-138-5p*, and *hsa-miR144-3p* were assessed in 26 traumatic brain injury (TBI) victims for 78 FFPE samples. For each case, the coup area, the contrecoup area, and the corpus callosum were analyzed. Sixty-two FFPE brain samples from forty-five subjects deceased due to acute cardiovascular events were used as control samples. Validated procedures were followed to assess these three miRNAs’ expression levels. In detail, the ∆∆Ct method was applied [23], with *hsa-miR-16-5p* and *hsa-miR-92-3p* as normalizing (housekeeping) markers.

As shown in Figure 4, the levels of the three miRNAs were significantly higher in the TBI samples (*p* < 0.01), with no differences among the coup area, the contrecoup area, and the corpus callosum. Thus, these results support that the three miRNAs selected here could be helpful in the diagnosis of TBI in forensic casework. In addition, the coup area and the surrounding areas show increased levels of these miRNAs. Although further experiments are needed to clarify the reasons for this finding, the alteration of these miRNAs in a large area of the brain agrees with the ability of miRNA molecules to be translocated into the extracellular department [10,14]. Nevertheless, diffusion from necrotic tissue is also likely [10,24].

The multivariate analysis confirmed the strong association between *hsa-miR-138-5p* and TBI (see Table 2). The role of this miRNA is not clear in the TBI samples studied here. This marker, in fact, is involved in glioma angiogenesis [20] and cancer cell proliferation [25]. In addition, it drives apoptosis [26]. Therefore, it is likely that *hsa-miR-138-5p* can be related to apoptosis even in TBI. Notably, the same statistical approach showed no correlation between the age at death (ranging from 23 to 90 years) and the dating of the FFPE samples (ranging from 2007 to 2021). This last finding implies that the age of the injured subject does not influence the expression levels; in addition, no bias seems to be introduced by FFPE tissue aging either.

When the expression levels of these three markers were compared with the agonal times, two miRNAs, namely, *hsa-miR-124-3p* and *hsa-miR-144-3p*, were shown to be significantly correlated (*p* < 0.01; see Table 2). In detail, *hsa-miR-124-3p* showed a positive correlation, which could agree with its role in neurogenesis [27]. It is of note that higher levels of this marker were selectively found in the plasma of severely injured patients [28]. On the contrary, *hsa-miR-144-3p* showed a negative correlation, which seems to agree with its role in the neuroinflammatory response [29]. *hsa-miRNA 144-5p* was already assessed as a candidate marker for use in grading the severity of trauma in TBI patients [30]; however, Di Pietro et al. did not detect any difference in the sera of patients with severe and mild TBIs [30]. Therefore, although the roles of *hsa-miR-124-3p* and *hsa-miR-144-3p* in the TBIs studied herein are not fully understood, the expression levels of these two markers could be potentially used even for timing a TBI. The results on *hsa-miR-138-5p*, which showed the most remarkable difference from the control samples (*p*-value < 0.00001; see Figure 1) even in the multivariate analysis (*p*-value: 0.004; see Table 1), returned no statistical association with the agonal time (*p*-value: 0.1). Noteworthily, no correlation was found between the expression levels of these three markers and the post-mortem intervals (PMIs). This last finding indicates the use potential of the three candidate markers, such as ideal tools in forensics for deciphering the cause and timing of death. 

The FFPE samples used in the present study had extensive levels of RNA degradation, as shown by the DV_100_ and DV_200_ values. RNA degradation is expected in FFPE tissues, especially tissues with autopsy origins. RNAase activity in the post-mortem phase [8] and chemical damage/modification due to extensive fixation [31,32,33] cause RNA degradation. Accordingly, the samples in our cohort with wider PMIs (up to 21 days) and longer fixation times had higher RNA degradation levels (see Table 2, Figure 2, and Appendix A). In addition, our control samples exhibited minor RNA degradation compared to the cases, as shown by the DV_200_ values (see Figure 1). The better preservation of the RNA molecules in the control sample is likely due to the shorter fixation periods and shorter PMIs. Although accurate data on the durations of fixation were not available, fixation was carried out for two days in 18 out of 45 (40.0%) control samples, whereas it lasted for up to several weeks in the TBI samples. This observation leads to the recommendation that standardized pre-analytical protocols must be established, even in forensics, if RNA analyses are planned. Molecular analyses call for higher-quality samples for reliable and reproducible results. Due to non-standardized procedures in autopsy, tissue mRNA profiling is hardly performed. In this work, the analysis of mRNA targets was not carried out because of the high levels of RNA degradation, a phenomenon that can produce artefactual results [34]. The analysis of miRNAs, even from highly degraded samples, has instead been proven to be more robust and reliable [35].

In conclusion, our work is the first study in which a large cohort (n = 26) of fatal TBIs were investigated using a set of candidate miRNAs. Although further studies are needed to validate the forensic use of the three miRNA markers studied here, our results indicate a strict correlation between their expression levels and TBIs. Lastly, *hsa-miR-124-3p* and *hsa-miR144-3p* have promise as suitable tools, even for dating TBIs. If biomarkers are planned to be routinely used in forensics, however, the definition of stringent pre-analytical procedures represents an irrefutable requisite for reliable and repeatable results [36].

## 4. Materials and Methods

### 4.1. Patients and Samples

A total of 78 paraffin-embedded tissue (FFPE) blocks from 26 brain trauma victims processed between 2007 and 2021 were retrieved from the Forensic Medicine Unit of the Azienda Sanitaria Universitaria Giuliano-Isontina (Trieste, Italy). 

The cohort included victims of traumatic brain injuries (TBIs) caused by road accidents, gunshots, falls, and assaults. For each case, three paraffin blocks were analyzed: the coup area, the contrecoup area, and the corpus callosum. When the corpus callosum was not available, the brainstem was collected. Two time variables, namely, the agonal time, representing the time between trauma and death, and the post-mortem interval, the time lapse between death and autopsy with sample collection, were gathered for TBI samples.

The cases were matched to controls by gender and age range for a total of 45 cases. For each control, at least one paraffin-embedded tissue block was collected from people who died from other causes (acute cardiovascular events) between 2007 and 2021 from the Forensic Medicine Unit of the Azienda Sanitaria Universitaria Giuliano-Isontina (Trieste, Italy) and of the Azienda Sanitaria Universitaria Friuli Occidentale (Udine, Italy).

### 4.2. Total RNA Extraction 

Four 10 μm thick sections were cut from micro-dissected FFPE tissue blocks, and the total RNA was isolated using a Maxwell^®^ RSC (Promega, Madison, WI, USA) device according to the manufacturer’s instructions, as already reported [37]. Briefly, the protocol was subdivided into two sections: the first section refers to the use of the Maxwell^®^ RSC RNA FFPE kit (Cat. No. AS1440, Promega, Madison, WI, USA), in which the sections were de-waxed, digested with proteinase K overnight, de-crosslinked, and treated with DNase I. The subsequent steps follow the Maxwell^®^ RSC miRNA Tissue Kit instructions (Cat. No. AS1460, Promega, Madison, WI, USA). The 1-Thioglycerol Homogenization solution and the Lysis Buffer (MC501C) were added to the aqueous phase. The entire lysate volume was added to the Maxwell^®^ RSC Cartridge and extracted via its specific method. The total RNA was eluted in 50 μL of nuclease-free water. Samples were split into aliquots and stored at −80 °C.

### 4.3. RNA Quantification and Quality

RNA concentration was measured using a Nanodrop ND 1000 spectrophotometer (Thermo Scientific, Waltham, MA, USA) using 1 μL of isolated RNA. 

The integrity of the RNA was assessed via microcapillary electrophoresis by submitting 1 μL of RNA to the Agilent 2100 Bioanalyzer (Agilent Technology, Santa Clara, CA, USA) through the use of the Agilent RNA 6000 nano kit. The total RNA quality was investigated via the analysis of the fragments. For each sample, three variables were gathered: the RIN (RNA Integrity number), the DV_100_, and the DV_200_, indicating the percentages of RNA fragments >100 nucleotides and >200 nucleotides, respectively.

### 4.4. Identification of miRNAs

The selection of miRNAs investigated in the present study was based firstly on a literature review in which miRNAs potentially related to neurodevelopment, neuro-regeneration, and neuronal injuries were selectively evaluated [10,19,20,21,38,39,40,41,42]. Accordingly, the following seven miRNAs were chosen: *hsa-miR-21-5p*, *hsa-miR-124-3p*, *hsa-miR-144-3p*, *hsa-miR-222-5p*, *hsa-miR-138-5p*, *hsa-miR-148b-5p*, and *hsa-miR-153-3p*, while the miRNAs *hsa-miR-16-5p*, *hsa-miR-101a-5p*, *hsa-let7a-5p*, *hsa-let7a-3p*, *hsa-miR-191-5p*, and *hsa-miR-92a-3p* were selected as calibrators [43,44,45]. In a second step, these miRNAs were tested on a subset of samples (N = 57), and those that were expressed at significantly different levels in cases and controls were analyzed in the entire cohort of the present study. The calibrator miRNAs, namely, *hsa-miR-16-5p*, *hsa-miR-let-7a*, *hsa-miR-191-5p*, and *hsa-miR-92a-3p*, were tested via a logistic regression analysis in a subgroup of selected samples (N = 36, 10 TBI deaths, including 23 specimens and 13 controls) according to their RIN values [46], to determine the most efficient, stable endogenous miRNAs with their detection limits.

### 4.5. Reverse Transcription and Real-Time PCR Assays of microRNAs

For the detection of the miRNAs, 7 ng of total RNA, as assessed by Nanodrop, was reverse-transcribed using a TaqMan^®^ Advanced miRNA cDNA Synthesis kit (Thermo Scientific, Waltham, MA, USA) according to the manufacturer’s protocol. 

A real-time PCR (RT-qPCR) was carried out using 11.7 pg of diluted cDNA TaqMan^®^ Fast Advanced Master Mix (2×) and TaqMan^®^ Advanced miRNA Assay (20×) in a total reaction volume of 20 µL. Each reaction was run in duplicate. A real-time PCR analysis was performed on a C1000TM Thermal Cycler, Bio-Rad CFX (Bio-Rad, Hercules, CA, USA), using the following cycle conditions: 95 °C for 20 s, 40 cycles of 95 °C for 30 s, and 60 °C for 1 min. 

For relative quantification, the ∆∆Ct method was applied [23,34] in which, as housekeeping, the geometric means of two miRNAs, *hsa-miR-16-5p* and *hsa-miR-92-3p*, were employed (see Table 3 for sequences and other details).

### 4.6. Statistical Analyses

Parametric or non-parametric tests were performed after checking the data distribution (D’Agostino–Pearson omnibus normality test and Shapiro–Wilk normality test). For normally distributed variables, a t-test or ANOVA was used. The association of the ratio of the investigated miRNAs and the categorical variables was analyzed via the Kruskal–Wallis test. The correlations for non-parametric variables were assessed using Spearman’s rank test. A logistic regression was carried out to test the dependences of the cause of death on the miRNA expression levels, year of the samples, and age of death. Pearson’s goodness-of-fit test was used as a post-estimation test.

Outliers were excluded from the box–whisker plot representations. Given the context of multiple testing, the significance level was set to α = 0.05/3 = 0.017 (Bonferroni correction). *p*-values ≤ 0.017 were considered statistically significant. Statistical analyses were carried out with the Stata/SE 16.0 package (StataCorp, College Station, TX, USA). 

## Figures and Tables

**Figure 1 ijms-24-10836-f001:**
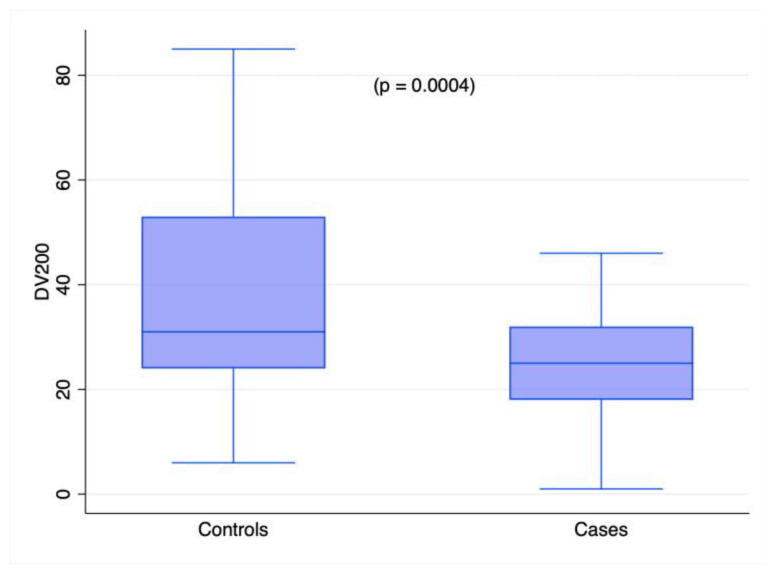
Box plot representing DV_200_ distribution between cases and controls. Kruskal–Wallis test.

**Figure 2 ijms-24-10836-f002:**
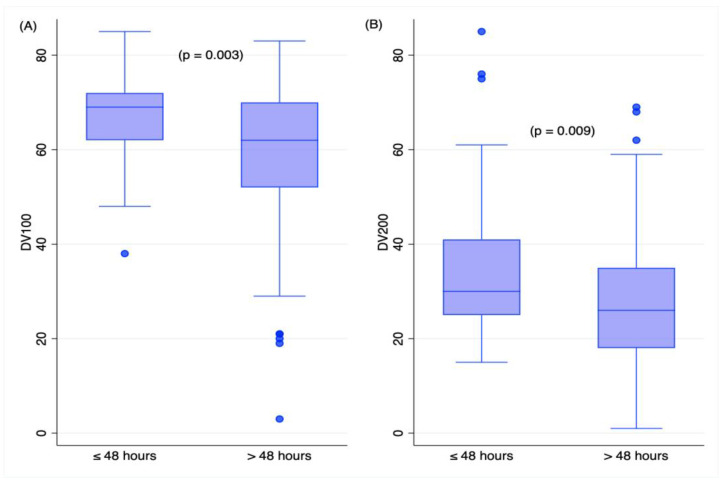
Box plot representing DV_100_ (**A**) and DV_200_ (**B**) according to fixation time. Student’s *t*-test.

**Figure 3 ijms-24-10836-f003:**
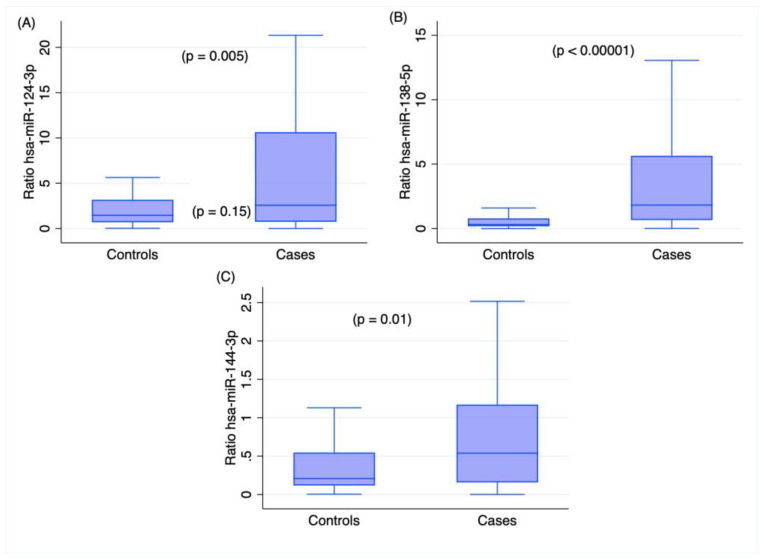
Box plot representing the expression levels of the analyzed miRNAs in cases and controls: (**A**) hsa-miR-124-3p, (**B**) hsa-miR-138-5p, and (**C**) hsa-miR-144-3p. Kruskal–Wallis test. Outliers were omitted from the graph.

**Figure 4 ijms-24-10836-f004:**
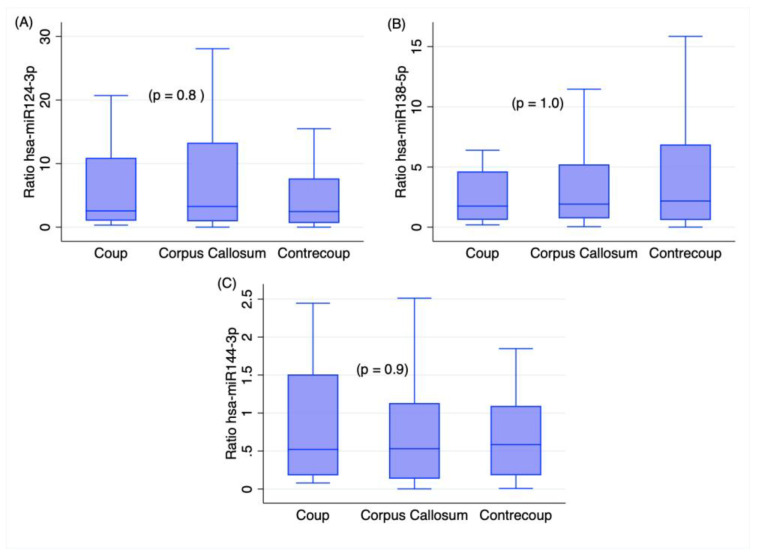
Box plot representing the expression levels of the analyzed miRNAs in the coup area, corpus callosum, and contrecoup area: (**A**) hsa-miR-124-3p, (**B**) hsa-miR-138-5p, and (**C**) hsa-miR-144-3p. Kruskal–Wallis test. Outliers were omitted from the graph.

**Table 1 ijms-24-10836-t001:** Results of the logistic regression.

Variables	Odds Ratio	*p* > |*z*|	95% Confidence Interval
*hsa-miR-124-3p*	1.02	0.5	0.96–1.10
*hsa-miR-138-5p*	1.62	0.004	1.17–2.25
*hsa-miR-144-3p*	1.09	0.2	0.94–1.26
Year of the sample	0.97	0.5	0.90–1.05
Age at death	1.02	0.3	0.99–1.05

**Table 2 ijms-24-10836-t002:** Results of Spearman’s rank correlation.

	Agonal Time		Post-Mortem Interval	
Variables	rho	*p* > |*t*|	rho	*p* > |*t*|
DV100	−0.14	0.5	−0.39	0.06
DV 200	0.018	0.9	−0.47	0.017
*hsa-miR-124-3p*	0.49	0.01	0.25	0.2
*hsa-miR-138-5p*	0.34	0.1	0.11	0.6
*hsa-miR-144-3p*	−0.53	0.007	−0.21	0.3

**Table 3 ijms-24-10836-t003:** microRNAs’ sequences and accession numbers and annealing temperatures used in PCR.

Name	Accession N ^1^	Sequence	Annealing T ^2^
*hsa-miR-16-5p*	MIMAT0000069	5′UAGCAGCACGUAAAUAUUGGCG	60 °C
*hsa-miR-92a-3p*	MIMAT0000092	5′UAUUGCACUUGUCCCGGCCUGU	60 °C
*hsa-miR-124-3p*	MIMAT0000422	5′UAAGGCACGCGGUGAAUGCCAA	60 °C
*hsa-miR-138-5p*	MIMAT0000430	5′AGCUGGUGUUGUGAAUCAGGCCG	60 °C
*hsa-miR-144-3p*	MIMAT0000436	5′UACAGUAUAGAUGAUGUACU	60 °C

^1^ N represents number; ^2^ T represents temperature.

## Data Availability

Data are reported in the Appendix A.

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
