# Peer review of "Evaluation of a Set of miRNAs in 26 Cases of Fatal Traumatic Brain Injuries"

_ijms, 2023, doi:10.3390/ijms241310836_

Round 1

Reviewer 1 Report

The manuscript is an original paper focused on the study of miRNAs in forensic cases of traumatic brain injuries (TBI). The authors picked up 3 out of 7 potential miRNAs implicated in neuronal differentiation and axonal growth; hypoxia and angiogenesis; intracerebral hemorrhage and neurodegeneration.

They checked first for RNA quality of the tissue samples which guarantees the reliability of the further analyses.

The number of cases included, the controls and the brain zones examined are correctly collected and processed. Are there any ethics issues (e.g. informed consent from relatives of deceased persons) considered?

In the Results section there is a technical omission – point 2.2 is either skipped or misnumbered.

The results show that the expression level of the analyzed miRNAs is independent from the sampling site of injured brains. The expression levels of the investigated miRNAs in TBI tissues and the indicators of RNA quality are related to agonal time. No association is detected between the post mortem interval and the expression of miRNAs.

The basic findings reveal that hsa-miR-124-3p showed a positive correlation with agonal time probably related to its role in neurogenesis.

hsa-miR-144-3p revealed a negative correlation possibly associated with  its role in neuroinflammatory response.

The originality of the work is in the investigation of a  set of candidate miRNAs in forensic cases which could be suitable even for dating the TBI.

Author Response

Q1 The number of cases included, the controls and the brain zones examined are correctly collected and processed. Are there any ethics issues (e.g. informed consent from relatives of deceased persons) considered?

A1 All the cases come from forensic autopsies and the possibility to use samples for scientific and diagnostic purposes is generally shared with relatives in this preliminary step from the prosecutor office. Anyway, this study has been approved from the local ethic committee. 

Q2 In the Results section there is a technical omission – point 2.2 is either skipped or misnumbered.

A2 Thank you for your kind advice. Point 2.2 was misnumbered effectively.

Reviewer 2 Report

The authors present a manuscript that identifies novel biomarkers(miRNAs) in cases of death from traumatic brain injury (TBI). Overall, the article needs a major revision for clarity of the work.

Introduction

-Line 35 Can you cite a more recent reference than #2?

Materials and Methods:

-Currently these are offered after the discussion on page 7 line 233. This is very confusing. Methods should follow the introduction. Overall the methods provide good details about sample identification, RNA and miRNA, and statistical analysis.

Results:

-the results are mixed with methods. It would be helpful if the results would be described in a results context.

Discussion:

-the discussion session comes following the results as appropriate.

-How does the TBI group compare to controls?

-Spelling and grammar edits are needed.

English language needs to be reviewed and edited.

Author Response

Comments made by the reviewer have been all valuable and very helpful for revising and improving our manuscript. We have studied the comments carefully and made corrections, which we hope meet with approval. The point-by-point response is included hereafter, and the revised portions are marked in the manuscript using the track changes system.

The authors present a manuscript that identifies novel biomarkers(miRNAs) in cases of death from traumatic brain injury (TBI). Overall, the article needs a major revision for clarity of the work.

Introduction

-Line 35 Can you cite a more recent reference than #2?

Response: Reference 2 is a milestone having 1533 citations.

Materials and Methods:

-Currently these are offered after the discussion on page 7 line 233. This is very confusing. Methods should follow the introduction. Overall the methods provide good details about sample identification, RNA and miRNA, and statistical analysis.

Response: We agree with the reviewer's comment; nevertheless, the journal guidelines foresee the material and method section as the last after the discussion.

Results:

-the results are mixed with methods. It would be helpful if the results would be described in a results context.

Response: The authors thank the reviewer for his/her helpful comment. Accordingly, some sentences were moved from the results to the material and method section.

Discussion:

-the discussion session comes following the results as appropriate.

-How does the TBI group compare to controls?

Response: The most appropriate statistical test was applied for comparison depending on the data distribution. Accordingly, the test was specified in the legend of the figures. Regarding the position of the discussion, as stated previously, we followed the journal’s guidelines.

-Spelling and grammar edits are needed.

Response: The authors thank the reviewer for this helpful comment. The manuscript was submitted for language revision.